# Digital inclusive finance penetration and household debt maturity structure: Empirical estimation based on China Family Panel Studies Data

**Jiayu Hou**[1]*, **Yingli Zhang**[2]

**1** College of Economics and Management, Shanghai Ocean University & Institute of Higher Education, Tongji University, Shanghai, China, **2** College of Economics and Management, Shanghai Ocean University, Shanghai, China

* shouhjy@yeah.net

## Abstract

This paper analyzes how the penetration of digital inclusive finance affects the maturity structure of household debt by matching data from the Peking University Digital Inclusive Finance Index and China Family Panel Studies (CFPS). Based on the conclusion, the broader penetration of digital inclusive finance has significantly improved the households' long-term debt level, thereby contributing to a longer-term trend in the maturity structure of household debt. Simultaneously, the effect is more evident in households with high income, large consumption expenditure, low asset levels, and no credit constraints, as well as households in central and western regions and those with high housing prices. According to the mechanism research, there are three transmission channels: the liquidity constraint of credit, household mental accounting, and the traditional bank competition. Therefore, we should focus not only on how digital inclusive finance impact on the overall scale of household debt, but also on how to create a reasonable debt maturity structure conducive to prevent financial risks.

## 1. Introduction

In recent years, with the improvement of the availability of financial resources, household debt has been used as the main tool to smooth household consumption. This has satisfied households' major asset purchases and inter-temporal consumption behaviors. As per studies, China's current household sector debt situation presents the following characteristics: the scale of household debt continues to rise, and the growth rate is significantly higher than the global average over the same time period; Long-term housing loans account for the major-ity of debt, which is also more distributed among households with low asset levels; whereas the debt level of the household in villages and towns is on average higher than that of urban households. The Financial Stability Report issued by the People's Bank of China pointed out that "there is an urgent need to regulate the high growth of household sector debt reasonably". At the same time, modern information technology innovations such as big data, 5G, artificial intelligence, and the Internet have promoted the vigorous development of digital inclusive

**Data availability statement:** The data used in our study were obtained from the China Family Panel Studies (CFPS). According to the CFPS user agreement, users are prohibited from sharing the entire CFPS dataset, partial datasets, or derivative datasets on third-party platforms. Consequently, the raw data involved in our study cannot be made publicly available under the CC BY 4.0. Fellow researchers seeking access to the CFPS data must follow the official application procedures, which include the following steps: 1. Visit the CFPS official website (https://www.isss.pku.edu.cn/cfps/en/index.htm) to review the data usage policies; 2. Submit a formal application through the CFPS official data portal (https://cfpsdata.pku.edu.cn/#/home), including personal information, institutional affiliation, research objectives, and a signed user agreement; 3. Upon approval, researchers will be granted access to the publicly available data via the CFPS official data portal (https://cfpsdata.pku.edu.cn/#/home).

**Funding:** The author(s) received no specific funding for this work.

**Competing interests:** The authors have declared that no competing interests exist.

finance represented by credit digitization and third-party payment. When it comes to meeting the needs of micro and small enterprises and low-asset-level households that usually have difficulty accessing traditional financial institutions' services, digital finance has emerged as a crucial power source and a new growth point of inclusive finance. Furthermore, it expands the scope of financial service regions and users, effectively alleviating financial exclusivity and improving the availability of financial resources. At the same time, emerging countries experiencing similar financial innovations can face problems such as adverse selection, loan access and increased defaults. Consumers tend to choose different borrowing channels based on their income, education level, and labor market risk, which not only affects their access to loans, but also has important implications for their repayment capacity and debt structure [1]. Moreover, after taking into account the impact of informal economic activities, the spread of digital payments, while contributing to economic growth, does not necessarily lead directly to an increase in total factor productivity (TFP), but rather indirectly supports economic development by increasing financial inclusion and credit availability [2,3]. However, as the digital credit business derived from digital inclusive finance has greatly improved the availability of credit for residents, it has aroused concerns about debt growth due to digital inclusive finance from all walks of life. Subsequently, how has the maturity structure of household debt altered and adjusted in light of the sharp rise of household debt promoted by the current boom in digital inclusive finance? Is there a severe hidden debt risk? In view of this, it is particularly crucial to explore the changes in the maturity structure of household debt and its potential risks in the context of booming digital financial inclusion.

The financial market has seen a significant rise in the importance of digital inclusive finance recently. This inclusiveness is considered one of the key characteristics of digital finance [4]. Digital inclusive finance emerged from the Internet as a tool that can satisfy those excluded from traditional finance, i.e., low-income and disadvantaged groups, demonstrating the essential characteristics of inclusive finance [5]. By utilizing the "digital footprint" accumulated by digital payments and identifying some high-quality groups excluded by traditional financial institutions, digital inclusive finance can enhance financial availability while reducing credit frictions such as credit rationing brought on by information asymmetry [6]. Concurrently, the ease of digital payments has allowed an increasing number of consumers access to multiple sources of funds and loans; while those without access to services in the formal financial sector can receive loans that to the "digital footprint" [7]. Unlike traditional finance, which physical outlets constrain, digital finance expands the reach of financial resources to include more "long-tail groups" to benefit from the digital dividend encouraged by the inclusive growth of digital technology. In such cases, digital finance, represented by online lending, can meet the demand for unsecured borrowing [8]. Higher debt leverage through digital finance channels may result from both the external bank financing market it creates and the strict monetary policies aimed at curbing leverage [9]. Thus, it is apparent that the process of "leveraging up" China's household sector has been significantly impacted by digital inclusive finance.

The rapid development of digital financial inclusion has had a profound impact on several economic sectors, not only changing the mode of operation of traditional financial institutions, but also bringing new changes to the financial behavior of households and individuals. Banking business processes and service models are undergoing significant transformation as digital financial tools become more widespread [10]. At the same time, through the use of Explainable Artificial Intelligence (EAI), financial institutions are able to better understand and serve their customers, thereby contributing to consumer upgrading. By improving the transparency and accessibility of financial services, EAI facilitates broader and deeper changes in consumer behavior [11]. In addition, the growing popularity of online trading platforms and mobile payment solutions has not only increased market activity, but also posed new

challenges to traditional investment analysis and risk management [12]. While digital finance presents many opportunities, we also need to be aware of its potential risks. While digital financial inclusion can help improve financial inclusion, it may also lead to some groups being more vulnerable to debt traps [13]. In this process, households with higher financial literacy are more effective in coping with financial crises and enhancing their financial resilience. This implies that improving the public's financial literacy is also one of the crucial tasks in promoting the development of digital financial inclusion [14]. Together, these studies reveal the complex impact of digital financial inclusion on household economic behavior and debt structure, which has undoubtedly become an integral part of the modern economic system.

Based on the above discussion, the research idea of this paper is to empirically estimate the effect of digital inclusive finance penetration on the maturity structure of household debt from the perspective of micro-finance, including whether this penetration has a boost and amplification effect on the maturity structure of short-term or long-term household debt. If so, what are the specific channels of effect transmission? In terms of research methodology, this paper matched the micro-sample data from the Peking University Digital Inclusive Finance Index and China Family Panel Studies (CFPS), then built a multi-dimensional panel fixed effect regression model to explore their internal relationship. Meanwhile, to ensure the robustness and reliability of the findings, this paper attempts to conduct robustness tests and heterogeneity analysis by employing various strategy methods. On this basis, this paper empirically investigates the specific channels of effect transmission. This study not only helps to deepen the understanding of how digital inclusive finance affects the economic behavior of households, but also provides important theoretical support and scientific basis for optimizing the design of related policies. Through systematic research on these issues, we hope to reveal the intrinsic links between digital inclusive finance and household debt and its maturity structure, and help it design more effective regulatory measures to prevent potential financial risks, thereby promoting healthy and stable economic and social development.

## 2. Literature review

### 2.1. Economic effects of digital inclusive finance

As the combination of digital Internet finance and financial services, digital inclusive finance has the advantages of easy payment, wide coverage, and low credit constraint. Hence, it is widely considered to positively impact inclusive socio-economic growth, business investment, household consumption, resident entrepreneurship and improvement of household financial structure. Based on data related to the Indian region from 1961–2002, it was noted that the production methods in rural areas changed with the development of the region's digital finance, while farmers' income increased with the development of digital finance [15]. Additionally, the positive effects of digital inclusive finance in expanding the scope of financial supply, improving the efficiency of financial supply, and reducing the transaction costs of financial markets have significantly improved household' participation in the financial market [16]. Concomitantly, households' involvement in the savings market, credit market, venture market, and Internet business activities is positively impacted by the development of digital inclusive finance [17]. Moreover, by offering real-time control over households' financial situations, the use of digital technologies to deliver new forms of financial services such as payments, lending, and investments has the potential to strengthen their balance sheets [18].

### 2.2. Factors influencing the structure of household debt

From the perspective of the debt supply side, the constraints on the credit market, the availability of financial products, the scope and accessibility of financial services, market interest

rates, Internet financial innovation, and other factors influence household debt structure. In contrast, property ownership status, financial asset ownership status, family age structure, risk preferences, expected income, social network, etc are among the debt demand side influences on household debt structure. Since the 1990s, the innovation of consumer loans and the liberalization of credit in the financial market have largely stimulated household credit availability and household liabilities growth [19]. Additionally, household debt in China is dominated by long-term housing loans, with a small fraction of short-term consumer loans [20]. Simultaneously, the credit market boom is the main contributor to household over-indebtedness [21]. Besides, the characteristics of digital finance include soft credit limitations, which can lead to over-crediting and over-borrowing in the credit market, thus expanding household debt along with increasing the vulnerability and instability of household indebtedness [22].

## 2.3. Review of the literature

Although existing research has laid a solid foundation for understanding the impact of digital financial inclusion on many aspects of the socio-economy, there are still some shortcomings. First, there is a relative lack of research at the micro level: most of the current research on the impact of digital financial inclusion focuses on macroeconomic growth and consumption levels, while there is less exploration of how it specifically affects the behavioral mechanisms of micro subjects such as households and individuals. In particular, research is scarce in the area of the impact of digital financial inclusion on household debt structure. This limits our understanding of how digital financial inclusion affects the maturity structure of household debt by altering household financial decisions. Second, there is a lack of in-depth exploration of the structure of household debt: while studies have pointed to the importance of the total size of household debt, not enough attention has been paid to the structural issues behind debt, particularly the specific mechanisms of how digital financial inclusion affects different types of debt (e.g., housing loans, consumer loans) and the maturity structure of these debts. This absence makes it difficult to fully assess the long-term impact of digital financial inclusion on household financial health. Third, the lack of heterogeneity analysis: most of the existing studies fail to adequately take into account the heterogeneity of differences in household backgrounds (e.g., age, income level, level of regional economic development, etc.) in terms of the acceptance of digital financial inclusion and its impact on household debt. This means that the existing findings may not accurately reflect the real situation of all types of households, especially those who are economically marginalized may have different response patterns. Fourth, the depth of the combination of theory and empirical evidence is insufficient: although some studies have begun to try to explore the impact of digital financial inclusion on household behavior, they often lack a strong theoretical framework to support their hypotheses or conclusions. In addition, in the empirical analysis, the difficulty of data acquisition, variable measurement errors and other problems lead to possible bias in the research results, which also requires that future research needs to pay more attention to the methodological innovation of combining theoretical construction and empirical verification.

Compared with prior literature, the potential marginal contributions of this study are as follows: Firstly, matching the panel data of the China Family Panel Studies and the Peking University Digital Inclusive Finance Index in 2014, 2016, 2018 and 2020 to explore the internal relationship between the penetration of digital inclusive finance and the maturity structure of household debt from a micro-empirical perspective. In this context, the robustness of the impact of digital inclusive financial penetration on the maturity structure of household debt is verified using different strategy methods. Secondly, while grasping the maturity structure of household debt, this paper analyzes the differential effects of different types of household

subgroups under the influence of digital inclusive finance. Last but not least, this paper analyzes and tests the effects of the liquidity constraints of credit, household mental accounting, and traditional bank competition on the maturity structure of household debt from the perspective of digital inclusive finance.

## 3. Theoretical mechanism

The penetration of digital inclusive finance has an important impact on both the demand and supply of household debt. However, the mechanism by which the penetration of digital finance affects the maturity structure of household debt is not clear [23]. By sorting out the existing literature, this paper examines the action mechanism from the following three possible aspects.

### 3.1. Liquidity constraint mechanism of credit

Financial innovation broadens consumer choice, improves household credit availability, and stimulates household debt growth [24]. By extending credit to individuals with insufficient credit, the rapid expansion of market loans may raise household debt and put borrowers in danger of an unsustainable debt trap [25]. Through information technology, digital financial development has reduced the degree of information asymmetry and the threshold of financial services, solving the long-standing issues of inadequate financial services and financial exclusion. As a result, a favorable credit policy environment which will stimulate households' willingness to borrow is created [26]. Concurrently, the inclusive performance of digital inclusive finance can rely on the data elements precipitated by digital payment to reduce information asymmetry in the financial market and relax the initial wealth constraints of institutions on individuals in need of capital. In addition, the "Macmillan Gap" is substantially closed by lowering the financial participation threshold, thus making loans available to marginalized groups excluded by traditional finance and reaching the "long tail groups". As a result, not only is it easier for vulnerable groups to obtain financial services at a relatively low cost, but it also improves financial accessibility and the "tail effect" of financial supply, easing capital constraints and broadening borrowing channels [27]. Therefore, the penetration of digital inclusive finance relieves the liquidity constraints of large and long-term household debts to a greater extent and in a wider range, allowing households to satisfy their large asset purchases and intertemporal consumption behaviors while lowering the level of long-term household debts.

### 3.2. Mechanism of family mental accounting

On the one hand, households experience a cognitive bias of income illusion in their subconscious due to the comparatively low borrowing interest rate of digital inclusive finance, which causes "mental accounting" and residents' continuous expansion of budget constraints [27]. On the contrary, digital inclusive finance continuously predicts individual spending preferences using the digital footprint generated by third-party payment and online consumer mall browsing records, exacerbating households' irrational spending. Furthermore, when microeconomic agents make economic decisions, these systems of accounts often follow certain underlying psychological arithmetic rules that contradict the laws of economic arithmetic. Regarding bookkeeping and behavioral decisions, these rules diverge significantly from rational economic and mathematical operations, thus they frequently have an unintended influence on individuals' economic decisions, leading them to make choices that violate the simplest economic principles and conduct irrational behavior [28]. Consequently, the "mental account effect" of households themselves causes them to often deviate from some basic economic algorithms, thus irrational

financial decisions occur. In this situation, individuals intend to maximize emotional satisfaction during mental operation rather than maximize utility in rational cognition. Furthermore, the "mental accounting" of households is worsened by the relatively low borrowing interest rates and lengthy maturities of digital inclusive finance, which further expand household budget constraints, thus stimulating households' irrational loan demand. As a consequence, with mindless spending, households' self-control in balancing large asset acquisitions and intertemporal consumption behavior may fail, and the demand for long-term borrowing leading to a rise in the overall level of long-term household debt.

### 3.3. Competition effect mechanism of traditional bank

Following the deep integration of finance with digitalization and inclusiveness, traditional commercial banks have gradually shifted from the initial single competitive relationship to cross-border competition and cooperation in response to the challenges of financial innovation companies. The four main forms of cooperation are acquisition, alliance, incubation, and joint venture [29]. While concurrently reducing potential competitors and increasing the market share of banks in the financial services sector, such cross-border competition and cooperation can lessen the degree of bank competition [30]. On this basis, the long tail effect of digital finance and the decreasing marginal cost effect have brought economies of scale to the traditional banking industry [31]. According to Marshall's Dilemma, the formation of economies of scale will inevitably raise the market share of products, which will ultimately result in market monopoly, and when monopoly develops to a certain extent, it will inevitably preclude competition. As digital inclusive finance overcomes the limitations of physical branches, financial resources expand to more "long-tail groups", which brings economies of scale to traditional commercial banks and increases the market share of banks' deposit and loan business, thus reducing the degree of competition within the banking industry. Driven by profit objectives, banks have increased the size and risk appetite of their loans, with a tendency to lend larger amounts with longer maturities to the target households following the reduced intensity of competition among banks owing to deepened monopoly of banks. Correspondingly, long-term bank loans raise long-term household debt since they are the most vital source of debt financing for households to acquire large assets.

## 4. Study design

### 4.1. Model setup

To verify the impact of digital inclusive finance penetration on the maturity structure of household debt, the following multi-dimensional panel fixed effect regression model is set for empirical estimation:

$$Debt_{i,t} = \beta_0 + \beta_1 DIFP_{i,n,t} + \beta_2 X_{i,t} + \beta_3 D_{i,t} + \tau_t + \gamma_i + \mu_{i,n,t}$$

In the model, $Debt_{i,t}$ represents the debt level of the ith household in year t; $DIFP_{i,n,t}$ represents the total index of digital financial inclusion in year t in the region n to which the ith household belongs; $X_{i,t}$ and $D_{i,t}$ represent the control variables at the household characteristics level and the regional macro level respectively; $\tau_t$ and $\gamma_i$ represent the time trend and the fixed effects of district and county, respectively; $\mu_{i,n,t}$ are the random disturbance terms.

### 4.2. Data source

The CFPS panel data in 2014, 2016, 2018 and 2020 and the Peking University Digital Inclusive Finance Index are matched at the prefecture-level as the sample set of empirical estimation.

Combining micro-level household and individual data with macro-level digital financial inclusion indices for multilevel data analysis helps us examine the impact of the macro environment on individual behavior while controlling for individual heterogeneity. For example, when studying the household debt maturity structure, we not only focus on factors such as the household's own financial situation and demographic characteristics, but also need to consider the impact of the external environment (e.g., the availability and penetration of digital financial inclusion services) on these decisions. In addition, using the panel data of CFPS, we can conduct a dynamic analysis to observe the adjustment of household debt maturity structure over time, and at the same time assess the impact of the development of digital inclusive finance on this adjustment process to enhance the explanatory power of the model. Numerous established studies have validated the effectiveness of the approach of combining micro-household data with macro digital financial inclusion indices, and this cross-level data matching helps to provide insights into how digital financial inclusion affects the economic activities of households by altering their financial behaviors [4,5,16–18,22,23,26–28].

Every two years, the CFPS database is followed up. 15000 households have been questioned, providing fundamental data on household finances such as household assets and liabilities, household demographic characteristics, and outstanding loans. These interviews took place in 162 counties throughout around 25 provinces.

Massive amounts of data on digital inclusive finance from Ant Financial Services were used to compile the Peking University Digital Inclusive Finance Index. The index includes several secondary subdivision indicators, such as the depth and coverage of digital inclusive financial services and degree of digital support services, can more comprehensively represent the impact of digital technology on household financial decision-making and reflect the inclusive and digital nature of digital inclusive finance.

### 4.3. Variable selection

**4.3.1. Maturity structure of household debt.** Explained variable: the household debt level, measured by the amount of debt to be repaid. According to the maturity structure, it can be divided into long-term debt level and short-term debt level. This paper uses the total amount of outstanding mortgage loans to measure the level of household long-term debt, primarily including loans obtained from commercial banks; Non-mortgage financial liabilities are used to measure the level of short-term household debt, which mainly includes Internet micro-consumer credit, private loans and loans from relatives and friends. "How much do you owe to the bank for the mortgage?" and "Do you owe money to other organizations or individuals (such as private credit institutions, relatives, friends, acquaintances, etc.) for reasons other than purchasing or building a house?" are the corresponding questions in the CFPS questionnaire. The use of total outstanding mortgage loans as a measure is mainly due to the fact that home purchase loans are usually the largest long-term liabilities of households and that they are large in amount and long in duration. Home purchase loans not only reflect a household's housing needs, but also its behavioral pattern of using credit for large asset acquisitions, and long-term debt, especially mortgage loans, has a significant impact on a household's financial health and stability [16–18,22,23,27].

**4.3.2. Penetration of digital inclusive finance.** Core explanatory variable: Penetration of digital inclusive finance. In this paper, the Peking University Digital Inclusive Finance Index is selected as the proxy variable to measure the penetration degree of digital inclusive finance. The reason for selection is mainly based on the fact that this index is the first research result covering the county level and describing the connotation and characteristics of digital finance penetration in multiple dimensions, which is more consistent with the research purpose of this paper. In this paper, the prefecture level index of this index is matched with

the geographical location of households in CFPS data to measure the spatial agglomeration and spatial heterogeneity of digital financial inclusion penetration in China. The use of this index allows for an effective assessment of the impact of digital technology diffusion on the structure of household debt, particularly in terms of enhancing the ability of households to access financial services [16–18,22,23,26–28].

**4.3.3. Control variables.** Referring to existing literature, this paper adds additional variables that affect the maturity level of household debt, mainly household characteristic variables to reduce the endogeneity problem caused by missing variables: family size, total household income, household consumption expenditure, level of household assets, and constraints on credit, the latter of which is constructed as a dummy variable using the CFPS questionnaire's question of "whether the household's loan has been rejected by banks or non-bank formal financial institutions" (if yes, credit constraint equals 1; if no, it equals 0); Regional macroeconomic variables: degree of urban-rural disparity in each province, the degree of marketization in each province. Lastly, this paper includes time trend fixed effects and regional district as well as county fixed effects to control the time trend and district and county differences in the level of household debt maturity.

Table 1 shows the sources, meanings and calculation methods of all the variables mentioned above.

## 4.4. Sample selection and data description

Throughout the data collation process, to alleviate the estimation bias caused by outliers, this paper preprocesses the data as follows: First, the missing values and invalid values are replaced or eliminated. For example, the value "-8" in the data indicates the missing caused by the leap of the questionnaire system, which was replaced by 0 according to the existing literature operation. Next, we winsorized the household long-term debt level, short-term debt level, total household income, consumption expenditure, and other economic indicators at levels above 1% and 99%. Table 2 provides comparable descriptive statistics of the regression variables in the entire sample.

**Table 1. Main variables and sources.**

| Name of variables | Source | Description |
|---|---|---|
| Explained variables: | | |
| Long-term debt level | CFPS | Log value of total outstanding mortgage loans |
| Short-term debt level | CFPS | Log value of financial liabilities other than mortgages |
| Explanatory variables: | | |
| Penetration of digital inclusive finance | IDF | China Digital Inclusive Finance Index |
| Control variables: | | |
| Family size | CFPS | Number of family members |
| Constraints on credit | CFPS | Whether there is a credit constraint (Yes = 1) |
| Total household income | CFPS | Log value of total household income |
| Household consumption expenditure | CFPS | Log value of household consumption expenditure |
| Level of household assets | CFPS | Log value of cash and savings |
| Degree of urban-rural disparity | CSMAR | Ratio of rural to urban per capita disposable income |
| Degree of marketization | CNBS | Marketization index of each province |

**Legend:** CFPS: China Family Panel Studies Database; IDF: Institute of Digital Finance, Peking University; CSMAR: China Stock Market & Accounting Research Database; CNBS: China National Bureau of Statistics.

**Table 2. Descriptive statistics of variables.**

| Name of variable | Observations | SD | Mean | Min | Max |
|---|---|---|---|---|---|
| Long-term debt level | 34838 | 0.7419 | 0.2109 | 0 | 3.7136 |
| Short-term debt level | 34562 | 4.0012 | 1.8564 | 0 | 12.2549 |
| Penetration of digital inclusive finance | 34838 | 42.8015 | 191.4069 | 105.6100 | 302.9800 |
| Family size | 34838 | 1.8945 | 3.8302 | 1 | 21 |
| Constraints on credit | 34694 | 0.4147 | 0.2207 | 0 | 1 |
| Total household income | 33807 | 1.8042 | 10.2617 | 0 | 12.8479 |
| Household consumption expenditure | 32525 | 0.9235 | 10.4510 | 8.0140 | 12.6785 |
| Level of household assets | 34774 | 5.0149 | 6.1319 | 0 | 15.6073 |
| Degree of urban-rural disparity | 34838 | 0.3996 | 2.6208 | 2.0350 | 3.4746 |
| Degree of marketization | 34838 | 1.7494 | 7.1500 | 3.8610 | 10.3700 |

*Notes*: The sample period covers all of 2014, 2016, 2018, 2020.

## 5. Empirical results

### 5.1. Baseline regression

Table 3 presents the benchmark estimation results of digital inclusive financial penetration on the maturity structure of household debt. All models account for time trends and regional district and county fixed effects. In accordance with Column (1), the regression coefficient for digital inclusive finance is 0.0018, which is significant at the 1% significance level. This shows that, when other conditions remain unchanged, every standard deviation increase in the penetration of digital inclusive finance leads to an average increase of 36.53%(0.0018*42.8015/0.2109) in the level of household long-term debt and explains 61.26% of the variation in its structure. On the contrary, Column (2) shows that, while other conditions stay constant, there is no discernible correlation between digital inclusive finance penetration and the level of household short-term debt. Columns (1) and (2) demonstrate that, when other conditions are fixed, the penetration of digital inclusive finance has no significant effect on the growth rate of the short-term maturity structure of household debt but only significantly influences the average level of the growth rate of the long-term maturity structure of household debt. With the deepening penetration of digital inclusive finance, looser long-term credit policies and bullish real estate asset prices may lead borrowers to be blindly optimistic, underestimate long-term borrowing risks and take on excessive long-term liabilities. Specifically, the penetration of digital inclusive finance significantly contributes to the trend of long-term household debt levels, while having no significant impact on the level of household debt for the short term, mainly due to, first, the increase in the richness and transparency of credit information. Digital inclusive finance is able to collect and analyze data on household and individual financial behavior more comprehensively through big data, artificial intelligence, and other technological means. This includes not only traditional income and expenditure records, but also covers a wide range of dimensions of information such as consumption habits and payment behavior. This wealth of data can help financial institutions better assess the creditworthiness of borrowers, thereby reducing credit risk. For example, Ant Gold Service's Sesame Credit Score is a credit evaluation system generated based on users' multifaceted behavioral data, which helps identify potential quality borrowers. Second, the specificity of the SMS lending market. Fintech companies usually focus on providing microfinance or short-term loans, mainly for individuals or micro and small enterprises who are in urgent need of financial liquidity and cannot obtain loans from traditional banks in a timely

manner. Such customers often have high liquidity needs, and while digital financial inclusion has improved their access to financial services, it has not fundamentally changed the pattern of their need for short-term funding. Even with better access to financial services, these clients still need frequent small loans to cover daily expenses or other emergencies. Moreover, fintech companies have designed their short-term loan products with the specific needs of their target customers and market gaps in mind. Digital financial inclusion penetration has not changed the core value proposition of these products, i.e., features such as fast approval and flexible repayment still appeal to specific types of borrowers. Therefore, in the short term, digital financial inclusion penetration will have little impact on the short-term lending business of fintech companies.

After controlling the household characteristic and regional economic variables, the estimation results in Table 3 Columns (3) and (4) indicate that the regression coefficient of digital inclusive financial penetration is still at the significance level of 5%, showing a significantly positive impact. This shows how the penetration of digital finance has significantly promoted the improvement of household long-term debt structure. Taking Column (4) as an example, from the quantitative point of view, every standard deviation increase in the penetration of digital inclusive finance will increase the structure of long-term household debt by 34.51% (0.0017*42.8015/0.2109), and accordingly explains 63.5% of the variation in the structure of long-term household debt. What's more, the regression results of the control variables of household characteristics show that under the influence of digital inclusive finance, the

**Table 3. Baseline regression results 1.**

| Explained variable | Short-term debt level | Long-term debt level | | |
|---|---|---|---|---|
| | (1) | (2) | (3) | (4) |
| Penetration of digital inclusive finance | -0.0011 (0.0041) | 0.0018*** (0.0007) | 0.0016** (0.0007) | 0.0017** (0.0007) |
| Family size | | | 0.3579*** (0.0048) | 0.3571*** (0.0048) |
| Constraints on credit | | | -0.0528*** (0.0116) | -0.0527*** (0.0115) |
| Total household income | | | -0.0010*** (0.0048) | -0.0010*** (0.0030) |
| Household consumption expenditure | | | 0.0597*** (0.0071) | 0.0597*** (0.0071) |
| Level of household assets | | | -0.0087*** (0.0010) | -0.0088*** (0.0010) |
| Degree of urban-rural disparity | | | | -0.3296** (0.1790) |
| Degree of marketization | | | | 0.0193 (0.0207) |
| Time fixed effect | YES | YES | YES | YES |
| Region fixed effect | YES | YES | YES | YES |
| $R^2$ | / | 0.6126 | 0.6345 | 0.6348 |
| Adj. $R^2$ | / | 0.3872 | 0.4115 | 0.411 |
| F test | / | 6.9690*** | 47.7120*** | 36.3120*** |
| Sample size | 31541 | 31799 | 28572 | 28572 |

Notes: Table 3 (1) and (2) report the regression results of the penetration of digital inclusive finance on the household short-term debt level and the household long-term debt level respectively. From Table 3 (3) - (4), the control variables at the household level and the regional level are added successively. This regression result includes time and region fixed effects. ***, **, * denote statistical significance at 1%, 5% and 10% levels, respectively. Standard errors are in parentheses.

smaller the credit constraint, the higher the total household income, the higher the household consumption expenditure, and the lower the household asset level, the more likely the long-term debt level of the household is to be increased. Also, the regression results of regional economic control variables indicate that long-term household debt may be higher in regions with a smaller urban-rural gap under the influence of digital inclusive finance.

Examining the impact of each sub-index on the level of long-term household debt. Through the aforementioned regression analysis, households' long-term debt structures are greatly improved by the penetration of digital finance. Nevertheless, we should use caution in light of this finding. To further test the reliability of the regression results of the benchmark model and further analyze which levels of penetration of digital inclusive finance promote the rise of the household long-term debt structure, we apply the three second-level sub-indicators of the Peking University Digital Inclusive Finance Index, the breadth of coverage (number of Alipay accounts, number of bound bank cards), depth of use (payment, credit, insurance, investment, credit investigation) and degree of digital support services (convenience, cost of financial services) to conduct regression analysis on the level of long-term household debt and test whether the significance of the total index in the benchmark regression changes significantly from the sign. Table 4 presents the test results of the three second-level sub-indicators. The results shit is shown that the coverage breadth, depth of use, or degree of digital support services all have a significant beneficial impact on the level of long-term household debt. However, compared to the benchmark model, this result has no notable change in significance or sign, with the impact of coverage breadth being the most crucial factor.

## 5.2. Robustness test

### 5.2.1. Replacement of the explained variable.
In this paper, new dependent variables are constructed to replace the explained variable in the benchmark regression using various measurement methods. The explained variable in the benchmark regression model -- the household long-term debt level, is created based on the total amount of the household's outstanding mortgage. In the robustness test, considering that the "household long-term gearing ratio" partially reflects the proportion of total household assets realized through

**Table 4. Baseline regression results 2.**

| Explained variable | Long-term debt level | | |
|---|---|---|---|
| | (1) | (2) | (3) |
| Breadth of coverage | 0.0022***<br>(0.0011) | | |
| Depth of use | | 0.0008*<br>(0.0005) | |
| Degree of digital support services | | | 0.0006**<br>(0.0356) |
| Variable of control | YES | YES | YES |
| Time fixed effect | YES | YES | YES |
| Region fixed effect | YES | YES | YES |
| $R^2$ | 0.6352 | 0.6253 | 0.6357 |
| $F$ test | 38.3000*** | 35.9100*** | 36.0400*** |
| Sample size | 29210 | 30659 | 28572 |

Notes: Table 4 reports the regression results of the three second-level sub-indexes of the penetration of digital inclusive finance on the household long-term debt level respectively. This regression result includes all the control variables, time and region fixed effects. ***, **, * denote statistical significance at 1%, 5% and 10% levels, respectively. Standard errors are in parentheses.

long-term liabilities, this paper defines the "household long-term gearing ratio" as the ratio of long-term liabilities to total household assets. The total household assets comprise land, real estate, financial assets, productive fixed assets, and durable consumer goods. The explained variable of the benchmark regression is replaced by the household long-term gearing ratio, after which the regression results of the penetration of digital inclusive finance and the segmentation index on the household long-term gearing ratio are investigated respectively. As shown in Table 5, the penetration of digital inclusive finance and the three second-level sub-indexes have a significantly positive effect on the household long-term gearing ratio, which remains consistent with the benchmark regression results.

**5.2.2. Replacement of the explanatory variables.** This paper uses distinct measurement methods to construct new independent variables to replace the explanatory variables of the benchmark regression. The explanatory variable of the benchmark regression model -- the penetration of digital inclusive finance, is based on the Peking University China Digital Inclusive Finance Index, which measures the development status and spatial layout of digital inclusive finance in China. In the robustness test, considering that the "number of fintech enterprises" reflects the development degree and spatial distribution of regional digital inclusive finance to a certain extent. This paper defines "regional fintech development" as the number of fintech enterprises at the prefecture level. It replaces the core explanatory variable of the benchmark regression with fintech development to examine the regression results of regional fintech development on the maturity structure of long-term household debt, as shown in Table 6 Column (1). At the significance level of 5%, regional fintech development has a significantly positive impact on the household long-term debt level, indicating that the regression analysis results will not change due to the replacement of independent variables.

**5.2.3. Removal of samples with specificity.** This paper deletes the sample size of the municipalities to conduct regression analysis again due to the significant economic particularity of the samples of the municipalities directly under the Central Government in China, as well as the maturity structure of household debt and the development level of digital

**Table 5. Robustness test 1: Replacement of the explained variable.**

| Explained variable | Household long-term gearing ratio | | | |
|---|---|---|---|---|
| | (1) | (2) | (3) | (4) |
| Penetration of digital inclusive finance | 0.0037***<br>(0.0001) | | | |
| Breadth of coverage | | 0.0003*<br>(0.0002) | | |
| Depth of use | | | 0.0002*<br>(0.0001) | |
| Degree in digital support services | | | | 0.0001***<br>(0.0001) |
| Variable of control | YES | YES | YES | YES |
| Time fixed effect | YES | YES | YES | YES |
| Region fixed effect | YES | YES | YES | YES |
| $R^2$ | 0.6574 | 0.6572 | 0.6571 | 0.6573 |
| $F$ test | 24.3465*** | 23.3898*** | 23.1606*** | 23.9443*** |
| Sample size | 14475 | 14475 | 14475 | 14475 |

Notes: Table 5 reports the regression results of the penetration of digital inclusive finance and its sub-index on the household long-term gearing ratio. The household long-term gearing ratio is calculated by the ratio of household long-term debt to total assets. ***, **, * denote statistical significance at 1%, 5% and 10% levels, respectively. Standard errors are in parentheses.

**Table 6. Robustness test 2: Replacement of the explanatory variable & Removal of special samples.**

| Explained variable | Long-term debt level | | | | |
|---|---|---|---|---|---|
| | (1) | (2) | (3) | (4) | (5) |
| Regional fintech development | 0.0214**<br>(0.0124) | | | | |
| Penetration of digital inclusive finance | | 0.0018**<br>(0.0008) | | | |
| Breadth of coverage | | | 0.0023**<br>(0.0011) | | |
| Depth of use | | | | 0.0010*<br>(0.0006) | |
| Degree in digital support services | | | | | 0.0006*<br>(0.0003) |
| Variable of control | YES | YES | YES | YES | YES |
| Time fixed effect | YES | YES | YES | YES | YES |
| Region fixed effect | YES | YES | YES | YES | YES |
| $R^2$ | 0.6371 | 0.6218 | 0.6218 | 0.6217 | 0.6217 |
| $F$ test | 35.2857*** | 33.3421*** | 33.2243*** | 33.0365*** | 33.1611*** |
| Sample size | 27865 | 26258 | 26258 | 26258 | 26258 |

Notes: Table 6 (1) reports the regression results of the regional fintech development on the household long-term debt level. From (2) to (4), the regression results of the penetration of digital inclusive finance on the level of household long-term debt after excluding the special samples of municipalities are reported. ***, **, * denote statistical significance at 1%, 5% and 10% levels, respectively. Standard errors are in parentheses.

inclusive finance in each municipality to a certain extent. The regression results are displayed in Table 6 Columns (2)-(5). After removing the particular sample size of the municipality, both the digital inclusive finance penetration and each segmentation index are consistent with the benchmark regression results. To sum up, the benchmark regression results are robust and reliable, as proved by the robustness tests of the benchmark regression results performed using three different strategy methods since the results remain unchanged.

**5.2.4. Endogeneity discussions.** This paper is the first to construct an instrumental variable called "evaluation", which captures the subjective value that family members place on accessing information from the mobile Internet, and thus measures the extent to which families perceive and value digital technology. The choice of this indicator as an instrumental variable stems from the consideration of the following factors: first, the penetration of digital financial inclusion relies on the expansion of digital technology, and it is easier to penetrate digital financial inclusion when households attach more importance to digital technology, so this variable is significantly positively correlated with the penetration of digital financial inclusion. Second, households' emphasis on digital technology has no direct effect on households' long-term debt decisions and is not significantly correlated with other endogenous variables and error terms. Therefore the evaluation instrumental variable satisfies the introduction principle. In this paper, we also chose to construct the second instrumental variable, "Bartik", by multiplying the observed value of the digital financial inclusion index in the lagged period with the first-order difference in the time series over the same period of time of the index, followed by a two-stage regression. Since changes in the debt maturity structure of households in a given municipality are unlikely to have an impact on the penetration of digital inclusive finance, it is also unlikely that there is a significant correlation between the level of digital inclusion penetration in the country as a whole and the data for households in a given municipality. Therefore the Bartik instrumental variable is valid. As shown in Table 7 Columns (1)–(2), both instrumental variables are significantly correlated with digital financial

**Table 7. Robustness test 3: Endogeneity Discussion.**

| Explained variable | Penetration of digital inclusive finance | | | Long-term debt level | |
|---|---|---|---|---|---|
| | **(1)**<br>**First stage regression** | | | **(2)**<br>**Second stage regression** | |
| *evaluation* | 0.7255***<br>(0.0824) | | Penetration of digital inclusive finance | 0.0218*<br>(0.0014) | |
| *Bartik* | | 0.1213***<br>(0.0145) | Penetration of digital inclusive finance | | 0.0605**<br>(0.0075) |
| Variable of control | YES | | Variable of control | YES | |
| Time fixed effect | YES | | Time fixed effect | YES | |
| Region fixed effect | YES | | Region fixed effect | YES | |
| | | | *Wald test* | 0.4839<br>(11.9826) | 0.0249<br>(5.0346) |
| $R^2$ | 0.7821 | 0.9692 | $R^2$ | 0.2974 | 0.5345 |
| Sample size | 30843 | 28572 | Sample size | 28572 | 28572 |

Notes: Table 7 reports the results of a two-stage regression of the instrumental variables "evaluation" and "Bartik" on Penetration of digital inclusive finance and Long-term debt level, to discuss whether the model is endogenous. This regression result includes all the control variables, time and region fixed effects. ***, **, * denote statistical significance at 1%, 5% and 10% levels, respectively. Standard errors are in parentheses.

inclusion penetration. Meanwhile, the coefficients of digital financial inclusion penetration as indicated by the instrumental variables are still significantly positive, only with reduced significance, and the p-value of the Wald test is insignificant at a value greater than 10%, proving that the instrumental variables do not directly affect the explanatory variables. The regression results with the introduction of instrumental variables are generally consistent with the benchmark model, verifying the reliability of the benchmark regression results.

## 5.3. Heterogeneity analysis

This research has already discussed in detail how the penetration of digital inclusive finance affects the maturity structure of household debt. Yet, it does not adequately address the issue of the differential effects of different types of groups under the influence of the penetration of digital inclusive finance, as well as the internal factors causing such disparities. Due to the heterogeneity of household finances and regional economic development, different types of households have distinct expenditure structures and credit constraints, and the level of long-term debt they can afford may also vary. To this end, this paper will analyze the heterogeneity on three levels: household financial differences, regional differences in household distribution, and regional differences in the household price level respectively.

**5.3.1. Examination from the perspective of household finances.** This paper selects "whether the household is higher than the average household total income, average consumption expenditure, average asset level and whether the household faces credit constraints" to construct dummy variables. The heterogeneity of different household debt maturity structures is tested by interacting with the digital inclusive financial index variables, where the dummy variables are assigned a value of 1 if the total income, consumption expenditure, and asset level of the sample households are higher than the sample mean, and 0 if the opposite is true. Table 8 Column (1) displays the regression results. For households in the sample below the average total household income, each standard deviation increase in the penetration of digital inclusive finance leads to an average increase of 32.47% (0.0158*42.8015/0.2109) in the level of long-term debt. Similarly, for households whose consumption expenditure is lower than the sample average, Column (2)'s interactive

**Table 8. Heterogeneity analysis 1: Examining From the perspective of household financial characteristics.**

| Explained variable | Long-term debt level | | | |
|---|---|---|---|---|
| | (1) | (2) | (3) | (4) |
| Digital Inclusive Financial index | 0.0015*** (0.0007) | 0.0016*** (0.0007) | 0.0021*** (0.0007) | 0.0017*** (0.0001) |
| Total household income ×index | 0.0001*** (0.0001) | | | |
| Household consumption expenditure×index | | 0.0004*** (0.0001) | | |
| Level of household assets×index | | | -0.0003*** (0.0001) | |
| Credit constraints×index | | | | -0.0003*** (0.0001) |
| Variable of control | YES | YES | YES | YES |
| Time fixed effect | YES | YES | YES | YES |
| Region fixed effect | YES | YES | YES | YES |
| $R^2$ | 0.6325 | 0.6231 | 0.6336 | 0.6378 |
| $F$ test | 36.7721*** | 34.7645*** | 32.6678*** | 36.6409*** |
| Sample size | 29210 | 30659 | 28601 | 28572 |

Notes: Table 8 reports the heterogeneity of households with different financial characteristics through interactive regression. The digital Financial inclusion index measures the penetration of digital financial inclusion, and the index refers to the digital financial inclusion index. ***, **, * denote statistical significance at 1%, 5% and 10% levels, respectively. Standard errors are in parentheses.

regression results for household consumption expenditure imply that for every one standard deviation increase in the penetration of digital inclusive finance, the level of long-term household debt increases on average by 40.58%. For households whose consumption expenditure is higher than the sample average, every standard deviation rise in the penetration of digital inclusive finance leads to a 38.56% increase in household long-term debt. According to the regression results by household asset level shown in Column (3), for every standard deviation increase in the penetration of digital inclusive finance, the long-term debt level of households with low asset level and high asset level increase on average by 42.62% and 36.53%, respectively. To address whether the households are faced with credit constraints, the regression results of Column (4) show that for every 1% increase in the penetration of digital inclusive finance, the long-term debt level of the households without credit constraints increases by 34.50% on average, and that of the households facing credit constraints increases by an average of 28.41%. From this observation, we can deduce that the deepening penetration of digital inclusive finance can alleviate the credit constraints faced by households, thereby indicating the strengthening of credit constraints may inhibit households from obtaining credit resources from formal financial institutions, which in turn limits the long-term rise of household debt. To sum up, under the influence of digital inclusive financial penetration, the debt maturity structure of different types of households exhibits obvious heterogeneity: with the further deepening of digital inclusive finance penetration, it primarily further improves the long-term debt structure of households with high household income, high household consumption expenditure, low household asset level, and no credit constraints, while it relatively slowly improves the long-term debt structure of households with low income, low household consumption expenditure, high asset level and credit constraints. This structural difference serves as a reminder to focus more on the maturity structure of household debt and prevent the penetration of digital inclusive finance from increasing risks and burdens to households with low asset levels.

### 5.3.2. Examination from the perspective of regional distribution of household affiliation.

To explore the moderating effect of the variability of the regional distribution to which the households belong on the benchmark regression results, the sample in this paper is divided into three groups of households in the eastern, central, and western provinces, depending on the province to which the households belong. In addition, multi-dimensional panel fixed effects are then estimated on the long-term debt levels of households respectively. The regression results are shown in Table 9 Columns (1)-(3). Among the household groups in the central provinces, the positive impact of digital inclusive financial penetration on the level of long-term household debt is the most significant, followed by the western provinces, and finally, the eastern provinces (0.00223 > 0.00172 > 0.00168), which indicates that the differential effect of long-term debt maturity structure of households mainly exists in the central and western regions. It can be seen that financial intermediaries loosen the initial wealth constraints on those in need of capital precisely because of the inclusiveness of digital inclusive finance. And to a considerable extent, lowering the financial participation threshold closes the McMillan Gap in the central and western regions by extending loans to disadvantaged groups excluded by traditional finance, covering the "long tail groups" and boosting the "tail effect" of financial supply in these areas.

### 5.3.3. Examination from the perspective of the regional house price.

The sample is divided into two groups, households in high house price areas (top 50%) and households in low house price areas (bottom 50%), according to the ranking of house price levels (using the real estate index of each province) in the province to which the households belong. Subsequently, multi-dimensional panel fixed effects are estimated separately for the long-term debt levels of households to explore the moderating effect of the variability of house price levels in the area where the households reside on the benchmark regression results. From Table 9 Columns (4) and (5), the regression results imply a significant positive effect of digital inclusive financial penetration on household long-term debt levels in high and low housing price areas. However, in terms of coefficients, the moderating effect of digital inclusion financial penetration to increase household long-term debt levels is stronger for households in high housing price areas (0.00214 > 0.00144). Given the rise in real estate prices in China over the past decade, the expectation of house value preservation and appreciation of households in areas with high housing prices may enhance their willingness to purchase real estate with long-term loans. At the same time, the penetration of digital inclusive finance has a greater

**Table 9. Heterogeneity analysis 2: Examining from the perspective of regional distribution and housing price level.**

| Explained variable | Long-term debt level | | | | |
|---|---|---|---|---|---|
| | The West | The Centre | The East | low housing prices | high housing prices |
| | (1) | (2) | (3) | (4) | (5) |
| Penetration of digital inclusive finance | 0.0017** | 0.0022* | 0.0016 | 0.0014* | 0.0021* |
| | (0.0007) | (0.0013) | (0.0019) | (0.0009) | (0.0009) |
| Variable of control | YES | YES | YES | YES | YES |
| Time fixed effect | YES | YES | YES | YES | YES |
| Region fixed effect | YES | YES | YES | YES | YES |
| $R^2$ | 0.6307 | 0.6308 | 0.6297 | 0.6850 | 0.6849 |
| F test | 10.1612*** | 12.4421*** | 16.6301*** | 25.7012*** | 9.19812*** |
| Sample size | 7456 | 9097 | 11678 | 16755 | 5448 |

Notes: Table 9 reports the heterogeneity of households with different regional distribution and regional housing price level through grouped regression. ***, **, * denote statistical significance at 1%, 5% and 10% levels, respectively. Standard errors are in parentheses.

impact on the long-term debt decisions of households in areas with high housing prices than those in low housing prices, which is another outcome of the penetration of digital inclusive finance in areas with high housing prices.

## 5.4. Analysis of impact mechanisms

In line with the benchmark regression analysis, the penetration of digital inclusive finance and the three secondary sub-indexes, namely the coverage breadth, the depth of use, and the degree of digital support services, can significantly improve the maturity structure of long-term household debt. "What is the mechanism of the impact of digital inclusive finance penetration on the maturity structure of long-term household debt?", merits further discussion. As mentioned in the theoretical mechanism analysis section above, this paper explains three transmission channels of digital inclusive finance penetration on the maturity structure of household debt: liquidity constraints of credit, mental accounting of households, and competition effect of traditional banks. In order to effectively verify the three transmission channels, firstly, this paper uses the penetration of digital inclusive finance to conduct multi-dimensional panel fixed effect regression on the proxy variables of the liquidity constraints of credit, the mental accounting of households, and the competition effect of traditional banks, respectively. It then estimates the internal relationship between the penetration of digital inclusive finance and the liquidity constraints of credit, the mental accounting of households and the competition effect of traditional banks. Secondly, the mechanisms through which liquidity constraints of credit, the mental accounting of households, and the competition effect of traditional banks influence the maturity structure of household debt are examined.

**5.4.1. Liquidity constraint mechanism of credit.** Through utilizing the CFPS questionnaire "whether banks and non-bank formal financial institutions have rejected the household's borrowing", this paper assigns a value of 1 to the credit constraint if it exists, otherwise, a value of 0 to construct a measure of the credit liquidity constraint. The regression test results in Table 10 Column (1) show that, in terms of quantity, the liquidity constraint of credit decreases by -1.06 on average for every one standard deviation increase in the penetration of digital inclusive finance, indicating that the liquidity constraint of credit can be significantly reduced by the deepening of the penetration of digital inclusive finance. This further demonstrates that when the penetration of digital inclusive finance becomes more widely adopted, it will improve the availability of credit resources for households by easing the transmission of the liquidity constraint mechanism of credit, thus increasing the level of long-term debt maturity of households.

**5.4.2. Household mental accounting mechanism.** This paper uses total consumption expenditure from the CFPS data as a proxy variable to measure the "mental accounting" of households [27], which mainly includes household expenditure on equipment and daily necessities, clothing and footwear, culture education and entertainment, food, rent, health care, welfare, transfer and other consumption-related expenses. Due to the relatively low borrowing interest rate of digital inclusive finance, households have a cognitive bias of income illusion in their subconscious, which causes the mental accounting effect and a budget constraint on residents' ongoing consumption growth. From the test results in Table 10 Column (2), in terms of quantity, the total consumption expenditure of the household increases by 1.01 on average for every one standard deviation increase in the penetration of digital inclusive finance. Ergo, indicating that the deepening of the penetration of digital inclusive finance can significantly increase the households' consumption expenditure and cause the households' budget constraint to continue expanding. This would in turn, stimulate an irrational long-term loan demand of the household, leading to an imbalance in their ability

**Table 10. Results of mechanism analysis.**

| Explained variable | Constraints on credit | Total consumption expenditure | HHI |
|---|---|---|---|
| | (1) | (2) | (3) |
| Penetration of digital inclusive finance | -0.0052*** | 0.0050** | 0.0012*** |
| | (0.0015) | (0.0026) | (0.0001) |
| Control variable | YES | YES | YES |
| Time fixed effect | YES | YES | YES |
| Region fixed effect | YES | YES | YES |
| $R^2$ | 0.5439 | 0.7511 | 0.9903 |
| $F$ test | 74.7813*** | 151.9178*** | 399.6591*** |
| Sample size | 28572 | 28572 | 28572 |

Notes: Table 10 reports the regression results of the three action transmission channels through which the penetration of digital inclusive finance affects the maturity structure of household debt, which are liquidity constraints, the effect of household mental account and the competition effect of traditional banks. Credit on constraint is the proxy variable of liquidity constraint mechanism of credit; total consumption expenditure is the proxy variable of household mental accounting mechanism; HHI refers to the Herfindahl-Hirschman Index and is the proxy variable of competition effect mechanism of traditional banks. ***, **, * denote statistical significance at 1%, 5% and 10% levels, respectively. Standard errors are in parentheses.

to control their spending on luxury consumer goods, large asset purchasing, intertemporal consumption behavior, and the increase in long-term debt.

**5.4.3. Traditional bank competition mechanism.** To measure the degree of bank competition within a region, this paper introduces the Herfindahl-Hirschman Index (HHI) of the banking industry, which reflects the structural characteristics of the banking industry based on physical outlets, and is a negative indicator of the degree of competition in the banking industry: The larger its value, the more concentrated the structure of the banking industry and the lower the degree of competition; the smaller its value, the more decentralized the structure of the banking industry is and the more competition there is. Based on the test results shown in Table 10 Column (3), in terms of quantity, the HHI rises by 24.35% on average for every one standard deviation increase in the penetration of digital inclusive finance, which indicates that the deepening of digital inclusive financial penetration significantly reduces the level of competition of traditional banks and increases the proportion of long-term bank loans as a source of debt financing for the household acquisition of large assets, thus raising the level of long-term household debt.

# 6. Conclusions and recommendations

## 6.1. Conclusion

Based on the CFPS data of 2014, 2016, and 2018, this paper studies the impact of digital inclusive financial penetration on the maturity structure of household debt at the micro-financial level. The main conclusions of this paper are as follows:

Generally speaking, households' long-term debt structure has greatly improved due to the penetration of digital inclusive finance. The coverage breadth, depth of use, and degree of digital support services are the three second-level sub-indexes that all have a significant positive impact on the level of long-term household debt. Concerning quantity, every one standard deviation increase in the penetration of digital inclusive finance leads to an average 34.50% increase in the long-term debt structure of households.

Second, the robustness test shows that even after the regression of various strategy methods, such as replacing the core explanatory variable, explanatory variable, and removing special samples, the effect of digital inclusive financial penetration on the rise in long-term household debt level is still valid.

Third, heterogeneity analysis reveals that from the perspective of household financial characteristics, the impact of digital inclusive finance penetration on the increase in household long-term debt level is more pronounced in households with high income, high consumption expenditure, low asset level, and no credit constraints; whereas from the perspective of regional household distribution and regional housing price level, the moderating effect of digital inclusive financial penetration on the long-term household debt structure is stronger in the central and western regions as well as high housing price regions.

Fourth, the analysis of the action mechanism shows that the penetration of digital inclusive finance promotes the growth of long-term household debt through three transmission channels: the liquidity constraints of credit, households' mental accounting, and the competition effect of traditional banks.

### 6.2. Recommendation

The findings of this paper show that the penetration of digital inclusive finance significantly increases the level of long-term household debt. On the one hand, the residential household leverage ratio in our country remains high, and there may be potential risks in the household debt structure; On the other hand, a variety of financial innovation activities, such as digital inclusive finance are constantly emerging in an endless stream, and the credit access threshold is gradually lowering. Therefore, the characteristics of the leverage term structure generated in the process of debt expansion in the residential sector are important factors affecting systemic financial risks. Thereby, following the continuous improvement of the availability of credit resources, an issue that the management department should focus on is how to improve the monitoring system of household debts of financial institutions and improve the risk assessment, prevention, and control mechanism. Based on the above analysis, the policy implications of this paper are as follows:

First, we should guide the benign development of digital inclusive finance, emphasize its inclusiveness, and pay particular attention to its differential effects on different groups. When formulating policies, the potential changes in residents' financial welfare caused by the penetration of digital inclusive finance should be carefully considered.

Second, we should encourage households to attend to grasping moderation and prevent the excessive and rapid growth of household debt levels. It is vital to actively minimize the risk of excessive long-term household leverage and protect the bottom line of zero systemic financial risks as too high long-term household debt will impact the country's financial stability in the long run.

Last but not least, the risk management of digital inclusive finance should be improved in addition to comprehending the dynamic trade-off between promoting the penetration of digital inclusive finance and averting financial risks. Although household debt is more concentrated in long-term debt, such as residential mortgages, the growth rate of short-term consumer loans is still noteworthy. Hence, as financial innovation progresses, it is important to pay attention to its impact on the overall amount of household debt and how to manage and prevent financial risks by having a reasonable debt maturity structure.

### Supporting information

**S1 Data.  Research data.**
(ZIP)

### Acknowledgments

I thank Professor Yingli Zhang, Shanghai Institute of International Finance, and School of Economics and Management, Shanghai Ocean University for their valuable comments and full support.

## Author contributions

**Conceptualization:** Yingli Zhang.

**Data curation:** Jiayu Hou.

**Formal analysis:** Jiayu Hou.

**Funding acquisition:** Yingli Zhang.

**Investigation:** Jiayu Hou.

**Methodology:** Jiayu Hou, Yingli Zhang.

**Resources:** Yingli Zhang.

**Software:** Jiayu Hou.

**Visualization:** Yingli Zhang.

**Writing – original draft:** Jiayu Hou.

**Writing – review & editing:** Jiayu Hou.

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
