## [Decision Letter · Decision Letter 0]

23 Dec 2024

PONE-D-24-27672Digital inclusive finance penetration and household debt maturity structure: Empirical estimation based on China Family Panel Studies DataPLOS ONE

Dear Dr. Hou,

Thank you for submitting your manuscript to PLOS ONE. After careful consideration, we feel that it has merit but does not fully meet PLOS ONE’s publication criteria as it currently stands. Therefore, we invite you to submit a revised version of the manuscript that addresses the points raised during the review process. Especially, reviewer 1 and 3 provide you with good suggestions to position the paper better. 

We look forward to receiving your revised manuscript.

Kind regards,

Wajid Khan

Academic Editor

PLOS ONE

Reviewers' comments:

Reviewer's Responses to Questions

**Comments to the Author**

1. Is the manuscript technically sound, and do the data support the conclusions?

Reviewer #1: Partly

Reviewer #2: Yes

Reviewer #3: Yes

2. Has the statistical analysis been performed appropriately and rigorously? 

Reviewer #1: No

Reviewer #2: Yes

Reviewer #3: Yes

3. Have the authors made all data underlying the findings in their manuscript fully available?

Reviewer #1: Yes

Reviewer #2: Yes

Reviewer #3: Yes

4. Is the manuscript presented in an intelligible fashion and written in standard English?

Reviewer #1: No

Reviewer #2: Yes

Reviewer #3: Yes

5. Review Comments to the Author

Reviewer #1: The manuscript explores the impacts of digital inclusive finance on the maturity structure of household debt. The topic is interesting. However, there are substantial shortcomings as follows:

1. In the first section, the authors should address the importance and why they investigated the research topic. Except for the current status of the research ground, some literature should also be addressed to outline the cutting-edge of this study. In the literature review section, more updated literature is encouraged to be added. As for the sections of introduction and literature review, the authors are encouraged to cite the following references to address the importance of this study and improve the discussions of the literature gap.

[1] Digital finance development and bank liquidity creation DOI: 10.1016/j.irfa.2023.102839

[2] Explainable artificial intelligence for digital finance and consumption upgrading DOI: 10.1016/j.frl.2023.104489

[3] Digital finance era: Will individual investors become better players? DOI: 10.1016/j.intfin.2024.101935

[4] The rise of digital finance: Financial inclusion or debt trap? DOI: 10.1016/j.frl.2021.102604

[5] Financial and household financial resilience DOI: 10.1016/j.frl.2024.105378

[6] Does the marginal child increase household debt? - Evidence from the new fertility policy in China DOI: 10.1016/j.irfa.2021.101870

2. Regarding the research data, the authors should address the rationale for matching individual data with macro-level data.

3. The selection of variables should offer theoretical or literature support.

4. Regarding the empirical results, the authors are suggested to verify them along with prior studies.

5. In addition to robustness and heterogeneity checks, the endogeneity also should be addressed.

6. The limitations of the study should be carefully addressed in the last section.

7. The authors are suggested to carefully proofread the manuscript before submission since there are substantial writing mistakes, typos, and similar mistakes.

Reviewer #2: Dear authors,

I think that your paper on household debt in China and digital finance using the Chinese is very well written. My suggestion is to include more references for household finance surveys from other emerging countries, such as:

https://doi.org/10.1016/j.jimonfin.2021.102455

https://doi.org/10.1186/s40854-023-00458-6

https://ideas.repec.org/p/bis/biswps/1196.html

Kind regards,

Reviewer #3: The paper is interesting and extends our knowledge in financial inclusion and mechanisms at household-level.

There are significant issues that need to be addressed by the authors to better position the paper.

1. In the first paragraph of the paper, the authors summarize the debt profile of households. This seems to point to strong long-term debt even to households with low-income levels and in the absence of digital financial inclusion. It is unclear, therefore, the important role played by digital financial inclusion.

2. The main evidence is reported in Table 3 - to suggest that digital financial inclusion is positively associated with long-term debt. The definition of long-term debt underscores the fact that this debt is provided by traditional banks rather than digital credit providers. A few questions can be answered by the authors to better position the paper:

a. Does digital financial inclusion make households more bankable to traditional banks? The authors can provide evidence on whether digital financial inclusion reduces the credit risk of households, thereby, making them potentially credible candidates to traditional banks.

b. Why is there no meaningful impact of digital financial inclusion on the primary credit (short-term) provided by fintech companies?

3. The authors use level of long-term debt. I suggest they include the results of changes in long-term debt. The changes can be informative in explaining how changes in the digital financial inclusion index may be reflected in debt choices of households.

4. I suggest that the authors define debt maturity as the proportion of either long-term debt or short-term debt on total debt. This proxy can be directly explained as representing the shifts in debt by households. As things stand now, a positive coefficient on the digital financial inclusion, does not necessarily imply a shift in the debt maturity to the long side.

5. Are there counties with CFPS data but no digital financial inclusion? It seems odd to me that, this may be the case. However, if it is the case, the authors can perform a test to tease out how digital financial inclusion more closely impact household debt decisions. Even in the absence of such sample, the authors should be able to split the existing sample to test high-level inclusion versus low-level inclusion on long-term maturity.

6. The authors should proofread the text to correct typos. I recommend that the authors use the free version of Grammarly.

6. PLOS authors have the option to publish the peer review history of their article (what does this mean? ). If published, this will include your full peer review and any attached files.

**Do you want your identity to be public for this peer review?** For information about this choice, including consent withdrawal, please see our Privacy Policy .

Reviewer #1: No

Reviewer #2: No

Reviewer #3: No

---

## [Author Response · Author response to Decision Letter 0]

5 Feb 2025

Response Letter

Dear Editors and Reviewers,

We appreciate the opportunity to revise our manuscript titled “Digital inclusive finance penetration and household debt maturity structure: Empirical estimation based on China Family Panel Studies Data” [PONE-D-24-27672] and are grateful for the insightful comments provided by the reviewers. Those comments are all valuable and very helpful for revising and improving our paper, as well as the important guiding significance to our researches.

In the following, we have provided detailed responses to each of the reviewers’ comments. Revised portion are marked in red underline font in the manuscript. Additionally, we have conducted a comprehensive and careful revision of the main manuscript. In this response letter, the reviewers’ comments are presented in italics, and our corresponding changes and additions to the manuscript are highlighted in red text. We have tried our best to make all the revisions clear, and we hope that the revised manuscript meets the requirements for publication.

Responses to Comments of Reviewer #1:

Comment 1: In the first section, the authors should address the importance and why they investigated the research topic. Except for the current status of the research ground, some literature should also be addressed to outline the cutting-edge of this study. In the literature review section, more updated literature is encouraged to be added. As for the sections of introduction and literature review, the authors are encouraged to cite the following references to address the importance of this study and improve the discussions of the literature gap.

[1] Digital finance development and bank liquidity creation DOI: 10.1016/j.irfa.2023.102839

[2] Explainable artificial intelligence for digital finance and consumption upgrading DOI: 10.1016/j.frl.2023.104489

[3] Digital finance era: Will individual investors become better players? DOI: 10.1016/j.intfin.2024.101935

[4] The rise of digital finance: Financial inclusion or debt trap? DOI: 10.1016/j.frl.2021.102604

[5] Financial and household financial resilience DOI: 10.1016/j.frl.2024.105378

[6] Does the marginal child increase household debt? - Evidence from the new fertility policy in China DOI: 10.1016/j.irfa.2021.101870

Response to Comment 1: Thank you for your questions and valuable suggestions. In the first section, we provide an in-depth analysis of the importance of and reasons for our study, which aims to deepen the understanding of how digital financial inclusion affects household economic behavior, and also provides important theoretical support and scientific basis for optimizing the design of related policies.

The revised objective now reads: The rapid development of digital financial inclusion has had a profound impact on several economic sectors, not only changing the mode of operation of traditional financial institutions, but also bringing new changes to the financial behavior of households and individuals. Banking business processes and service models are undergoing significant transformation as digital financial tools become more widespread [10]. At the same time, through the use of Explainable Artificial Intelligence (EAI), financial institutions are able to better understand and serve their customers, thereby contributing to consumer upgrading. By improving the transparency and accessibility of financial services, EAI facilitates broader and deeper changes in consumer behavior [11]. In addition, the growing popularity of online trading platforms and mobile payment solutions has not only increased market activity, but also posed new challenges to traditional investment analysis and risk management [12]. While digital finance presents many opportunities, we also need to be aware of its potential risks. While digital financial inclusion can help improve financial inclusion, it may also lead to some groups being more vulnerable to debt traps [13]. In this process, households with higher financial literacy are more effective in coping with financial crises and enhancing their financial resilience. This implies that improving the public's financial literacy is also one of the crucial tasks in promoting the development of digital financial inclusion [14]. Together, these studies reveal the complex impact of digital financial inclusion on household economic behavior and debt structure, which has undoubtedly become an integral part of the modern economic system.

Based on the above discussion, the research idea of this paper is to empirically estimate the effect of digital inclusive finance penetration on the maturity structure of household debt from the perspective of micro-finance, including whether this penetration has a boost and amplification effect on the maturity structure of short-term or long-term household debt. If so, what are the specific channels of effect transmission? In terms of research methodology, this paper matched the micro-sample data from the Peking University Digital Inclusive Finance Index and China Family Panel Studies (CFPS) , then built a multi-dimensional panel fixed effect regression model to explore their internal relationship. Meanwhile, to ensure the robustness and reliability of the findings, this paper attempts to conduct robustness tests and heterogeneity analysis by employing various strategy methods. On this basis, this paper empirically investigates the specific channels of effect transmission. This study not only helps to deepen the understanding of how digital inclusive finance affects the economic behavior of households, but also provides important theoretical support and scientific basis for optimizing the design of related policies. Through systematic research on these issues, we hope to reveal the intrinsic links between digital inclusive finance and household debt and its maturity structure, and help it design more effective regulatory measures to prevent potential financial risks, thereby promoting healthy and stable economic and social development. (red underlined font on pages 2 and 3 in Revised Manuscript with Track Changes)

Comment 2: Regarding the research data, the authors should address the rationale for matching individual data with macro-level data.

Response to Comment 2: We sincerely appreciate the valuable comment. from an econometric point of view, combining micro-level household and individual data with macro-level digital financial inclusion indices, the reason:

(1) Multilevel Analysis can be performed to enhance the explanatory power of the model.

(2) In the presence of potential endogeneity problems, an instrumental variables approach can be naturally employed to address them. For example, the Digital Inclusion Index may be associated with certain unobservable household characteristics, leading to endogeneity problems. By finding suitable instrumental variables, it can help identify the true causal effect of digital financial inclusion on the term structure of household debt.

(3) Matching individual data with macro data is already a common practice in the established authoritative literature, for example:

[1]Guo F, Wang J Y, Wang F. Measuring the development of digital inclusive finance in China: index compilation and spatial characteristics. China Economic Quarterly. 2020; 19(04):1401-1418.

[2]Wu Z H, Ye J, Guo X H. Income inequality, social security expenditure and household borrowing behavior: an empirical analysis based on CFPS data. Journal of Financial Science and Economics. 2017; 17(12):55-68.

[3]Chen C, Fang F, Zhang L. Digital Inclusive Finance, Income Level and Household Indebtedness. Economic Survey. 2022; 22(01):127-137.

[4]Wang, H Y, Yue H. Li W Q. How does the development of digital finance affect households’ leveraging up? Dynamic effect, heterogeneity and mechanism test. South China Journal of Economics, 2021; 21(09):18-35.

The revised objective now reads: The CFPS panel data in 2014, 2016, 2018 and 2020 and the Peking University Digital Inclusive Finance Index are matched at the prefecture-level as the sample set of empirical estimation. Combining micro-level household and individual data with macro-level digital financial inclusion indices for multilevel data analysis helps us examine the impact of the macro environment on individual behavior while controlling for individual heterogeneity. For example, when studying the household debt maturity structure, we not only focus on factors such as the household's own financial situation and demographic characteristics, but also need to consider the impact of the external environment (e.g., the availability and penetration of digital financial inclusion services) on these decisions. In addition, using the panel data of CFPS, we can conduct a dynamic analysis to observe the adjustment of household debt maturity structure over time, and at the same time assess the impact of the development of digital inclusive finance on this adjustment process to enhance the explanatory power of the model. Numerous established studies have validated the effectiveness of the approach of combining micro-household data with macro digital financial inclusion indices, and this cross-level data matching helps to provide insights into how digital financial inclusion affects the economic activities of households by altering their financial behaviors [4,5,16-18,22,23,26-28]. (red underlined font on pages 6 in Revised Manuscript with Track Changes)

Comment 3: The selection of variables should offer theoretical or literature support.

Response to Comment 3: Thank you for your helpful suggestions. In order to enhance the rationality and scientific validity of these variable choices, we have listed in detail the definitions of all the variables, the data sources and their theoretical rationale, and further elaborated in the Methods section why these particular variables were chosen to test our hypotheses:

(1) Explained variables: household debt level

a. Long-term debt level: the total amount of outstanding home loans is used to measure this, mainly because home purchase loans are usually the largest long-term liabilities of households and they are large in amount and long in maturity (Wang et al., 2021; Chen et al., 2022). Home purchase loans not only reflect households' housing needs, but also their behavioral patterns of using credit for large asset acquisitions. Literature support suggests that long-term debt, especially home loans, has a significant impact on household financial health and stability.

b. Level of short-term debt: Measured by non-mortgage financial liabilities, including small consumer credit over the Internet, loans from private individuals and friends and relatives. According to Madeira (2023), consumers choose different types of loans to cope with liquidity needs or emergencies when they face different financial pressures. This division helps to understand how households choose different types of credit products for different needs, especially in the context of digital financial inclusion, where Internet microfinance has become an important source for meeting short-term financial needs.

(2) Core explanatory variable: digital financial inclusion index

The digital financial inclusion index covers multiple dimensions such as breadth of coverage, depth of use and digital support services (Aguilar et al., 2024), which can comprehensively reflect the popularity of digital financial services and the level of technology application within a region. It has been shown that the development of digital financial inclusion can help improve the accessibility of financial services, reduce transaction costs, and promote the participation of households in financial market activities (Yang et al., 2020; Zeng et al., 2022). Therefore, the use of this index can effectively assess the impact of digital technological advances on the structure of household debt, especially in terms of enhancing the ability of households to access financial services.

(3) Control variables:

a. Household Characteristics Variables: these include household size, total income, consumer spending, asset level and credit constraints. The selection of these variables is based on existing literature, such as He et al. (2010) and Zhuang et al. (2022), which indicate that the demographic characteristics, income level, and asset-liability status of a household have a direct impact on its borrowing behavior. In particular, the credit constraint indicator, which is constructed as a dummy variable through questions in the CFPS questionnaire, directly reflects the barriers that households encounter when attempting to access loans from formal financial institutions, which is critical to understanding why households turn to informal borrowing channels.

b. Regional macroeconomic variables: the degree of urban-rural disparity in each province, the degree of marketization, and the level of commodity housing prices are also included in the model as control variables. These factors affect households' borrowing capacity and preferences (Deng & Yu, 2022). For example, higher commodity housing prices may prompt more households to rely on loans to purchase homes, while the degree of marketization in a region may affect households' acceptance of different types of financial products.

c. Time-trend fixed effects and geographic district fixed effects: these effects are included to control for potential time-series correlation and geographic variation and to ensure that the estimates are not affected by unobserved time- or space-specific factors. This approach is commonly used in panel data analysis and aims to improve estimation precision and reduce bias (Yi et al., 2018; Yin et al., 2021).

The revised objective now reads: Explained variable: the household debt level, measured by the amount of debt to be repaid. According to the maturity structure, it can be divided into long-term debt level and short-term debt level. This paper uses the total amount of outstanding mortgage loans to measure the level of household long-term debt, primarily including loans obtained from commercial banks; Non-mortgage financial liabilities are used to measure the level of short-term household debt, which mainly includes Internet micro-consumer credit, private loans and loans from relatives and friends. “How much do you owe to the bank for the mortgage?” and “Do you owe money to other organizations or individuals (such as private credit institutions, relatives, friends, acquaintances, etc.) for reasons other than purchasing or building a house?” are the corresponding questions in the CFPS questionnaire. The use of total outstanding mortgage loans as a measure is mainly due to the fact that home purchase loans are usually the largest long-term liabilities of households and that they are large in amount and long in duration. Home purchase loans not only reflect a household's housing needs, but also its behavioral pattern of using credit for large asset acquisitions, and long-term debt, especially mortgage loans, has a significant impact on a household's financial health and stability [16-18,22,23,27].

Core explanatory variable: Penetration of digital inclusive finance. In this paper, the Peking University Digital Inclusive Finance Index is selected as the proxy variable to measure the penetration degree of digital inclusive finance. The reason for selection is mainly based on the fact that this index is the first research result covering the county level and describing the connotation and characteristics of digital finance penetration in multiple dimensions, which is more consistent with the research purpose of this paper. In this paper, the prefecture level index of this index is matched with the geographical location of households in CFPS data to measure the spatial agglomeration and spatial heterogeneity of digital financial inclusion penetration in China. The use of this index allows for an effective assessment of the impact of digital technology diffusion on the structure of household debt, particularly in terms of enhancing the ability of households to access financial services [16-18,22,23,26-28]. (red underlined font on pages 7 in Revised Manuscript with Track Changes)

Comment 4: Regarding the empirical results, the authors are suggested to verify them along with prior studies.

Re

---

## [Editor Report · Decision Letter 1]

13 Feb 2025

Digital inclusive finance penetration and household debt maturity structure: Empirical estimation based on China Family Panel Studies Data

PONE-D-24-27672R1

Dear Dr. Hou,

We’re pleased to inform you that your manuscript has been judged scientifically suitable for publication and will be formally accepted for publication once it meets all outstanding technical requirements.

Kind regards,

Jim Been

Academic Editor

PLOS ONE
---

## [Editor Report · Acceptance letter]

PONE-D-24-27672R1

PLOS ONE

Dear Dr. Hou,

I'm pleased to inform you that your manuscript has been deemed suitable for publication in PLOS ONE. Congratulations! Your manuscript is now being handed over to our production team.

Kind regards,

on behalf of

Dr. Jim Been

Academic Editor

PLOS ONE